# Genetic Analysis of Novel Behaviour Traits in Pigs Derived from Social Network Analysis

**DOI:** 10.3390/genes13040561

**Published:** 2022-03-23

**Authors:** Saif Agha, Simone Foister, Rainer Roehe, Simon P. Turner, Andrea Doeschl-Wilson

**Affiliations:** 1The Roslin Institute, University of Edinburgh, Easter Bush, Edinburgh EH25 9RG, UK; andrea.wilson@roslin.ed.ac.uk; 2Animal Production Department, Faculty of Agriculture, Ain Shams University, Cairo 11241, Egypt; 3Animal and Veterinary Sciences Department, Scotland’s Rural College, West Mains Road, Edinburgh EH25 9RG, UK; sf@itlscotland.co.uk (S.F.); rainer.roehe@sruc.ac.uk (R.R.); simon.turner@sruc.ac.uk (S.P.T.)

**Keywords:** social network analysis, genetic parameters, welfare, aggressiveness, pigs

## Abstract

Social network analysis (SNA) has provided novel traits that describe the role of individual pigs in aggression. The objectives were to (1) estimate the genetic parameters for these SNA traits, (2) quantify the genetic association between SNA and skin lesion traits, and (3) investigate the possible response to selection for SNA traits on skin lesion traits. Pigs were video recorded for 24 h post-mixing. The observed fight and bullying behaviour of each animal was used as input for the SNA. Skin lesions were counted on different body parts at 24 h (SL24h) and 3 weeks (SL3wk) post-mixing. A Bayesian approach estimated the genetic parameters of SNA traits and their association with skin lesions. SNA traits were heritable (h^2^ = 0.09 to 0.26) and strongly genetically correlated (rg > 0.88). Positive genetic correlations were observed between all SNA traits and anterior SL24h, except for clustering coefficient. Our results suggest that selection for an index that combines the eigenvector centrality and clustering coefficient could potentially decrease SL24h and SL3wk compared to selection for each trait separately. This study provides a first step towards potential integration of SNA traits into a multi-trait selection index for improving pigs’ welfare.

## 1. Introduction

Grouping of unacquainted pigs is a common procedure in commercial farms; however, it results in elevated aggression as animals attempt to establish stable dominance relationships [1]. Aggression reduces performance, health and welfare of pigs, and impacts negatively on farm efficiency and net-zero agriculture [2,3]. Despite the efforts made to find a practical solution to control aggression at mixing, this welfare issue is still unresolved on commercial farms [4]. Genetic selection could help in providing a long-term solution for controlling aggression [5]. The starting point for any genetic improvement program is to identify the desirable traits to be targeted for genetic improvement and determine the expected response to selection in these traits to achieve the breeding goals.

Several traits have been recommended as indicators of aggressive behaviour in pigs [6]. Among these traits, skin lesion counts can be considered as a practical proxy for aggression [7]. The moderate heritability and positive genetic correlation of skin lesions with several aggressive behaviour traits make them a valuable candidate for selective breeding [8,9,10]. However, it is important to consider the time and location at which skin lesions occur on the body, as they occur at different stages post-mixing and indicate different types of aggressive behaviour. For example, skin lesions to the anterior part of the body are received primarily from reciprocal aggression, whilst injuries of the rump often occur during retreat from non-reciprocated bullying [11]. In this context, Desire et al. (2016) [12] have considered skin lesion scores on different body regions (anterior, central, and posterior) at 24 h (SL24h) and 3 weeks (SL3wk) post-mixing as potential selection candidate traits for reducing aggressive behaviour in pigs. Their results suggested that selection against anterior SL24h could effectively reduce aggressive behaviour at mixing. However, subsequent studies showed that reducing lesions at mixing is unlikely to reduce lesions from chronic aggression [13].

Whilst skin lesion counts provide some indication of pigs’ aggressive behaviour, a significant part of the variance in aggressive interactions is not captured by the number of skin lesions received by individuals [13]. Most previous efforts to quantify skin lesions or aggressive interactions as potential selection traits have ignored the identity of opponents and assumed that dyadic contests occur independently of the wider structure of social relationships in the group [14]. This fails to recognise the opponent identity, previous experience with the same opponent and the previous interactions of both contestants with other group members, which are likely to inform aggressive behaviour and resulting lesions. Therefore, in order to reduce aggressive behaviour on a pen level it is important to understand and quantify the complex social structure within the group and identify the individuals that contribute most to aggressive behaviour in the pen.

Recently, social network analysis (SNA) of animal behaviour has gained much attention, due to its potential to provide novel insights into the social relationships between individuals for use in the fields of animal welfare science and fundamental ethology [15,16]. In pigs, SNA has been used to identify the dominance structure within the group and the indirect relationships between animals that are not captured by traditionally used dyadic traits [17]. Furthermore, the social network properties derived by SNA have provided a more accurate prediction of skin lesions of pigs compared to dyadic traits [18]. In addition to quantifying the social structure in a group, SNA also provides quantitative ‘centrality traits’ that describe the position of each animal in the social structure [19,20]. These individual SNA traits have the potential to identify the individuals that have a high influence on the short- and long-term stability of the social relationships in their pen, which indirectly affect the injury rates of all the animals in the group [18,21]. Only a few studies have attempted to estimate the genetic basis of network centrality in different species [22,23,24,25]. However, the existence of a genetic component contributing to aggressive network centrality in pigs is still unknown. Therefore, the objectives of this study were: (1) to estimate the genetic parameters for SNA centrality traits that have been identified to influence pen level aggressive behaviour and subsequent skin lesions either immediately post-mixing or in stable social conditions in Foister et al. (2018) [18]; (2) to quantify the genetic association between SNA and skin lesion traits; (3) to investigate the possible response to selection for network centrality on skin lesion traits based on estimated breeding values (EBVs) of the SNA traits at individual and pen levels.

## 2. Materials and Methods

### 2.1. Animals

This study uses the same dataset as described in Desire et al. (2016) and Foister et al. (2018) [12,18], which used dyadic and SNA traits, respectively, to assess aggressive behaviour associated with skin lesions in pigs. Briefly, the dataset consisted of 1146 commercial pigs, 698 of which were purebred Yorkshire and 448 were Yorkshire × Landrace on a commercial farm in Ransta, Sweden, between October 2005 and January 2007. They were housed in the same building with each ‘batch’ of animals (i.e., those on the experiment at the same moment in time) housed in the same room of this building. The building had automatically controlled natural ventilation with the temperature maned within the thermoneutral zone and adjusted as the pigs aged. Pigs were kept in 4.0 × 3.2 m partially slatted pens (30% slats, 70% lightly bedded solid flooring) with a floor space allowance of 0.85 m^2^ per pig. Animals were fed dry pelleted food ad libitum from a single space feeder and had constant access to water via a nipple drinker. The animals were the progeny of 82 sires and 217 dams. The total number of animals in the two-generation pedigree was 2427. Animals were distributed into 78 pens, i.e., social groups, each containing 15 animals of the same sex and breed which were formed using 3 animals from each of 5 litters. Animal weights were considered while grouping in order to minimize weight variation within the pen. The average age at mixing of the animals was 71 days (SD 4.5).

### 2.2. Skin Lesion Traits

Fresh skin lesions were counted for each body region 24 h post-mixing. One uninterrupted scratch was classed as a single lesion, regardless of length or severity. The body regions were separated into anterior (head, neck, forelegs, and shoulders), central (flanks and back) and posterior (hind legs and rump). To avoid including the injuries that occurred before mixing, skin lesions were counted immediately before mixing the animals and subtracted from the number counted 24 h post-mixing. Moreover, fresh lesions were counted again 3 weeks post-mixing as a measure of aggression under stable social conditions. A lesion was considered to be recent if it was vivid red in colour or recently scabbed. The skin lesion records were log transformed (y = loge (x + 1)) to approach the normal distribution. For further detailed information see Desire et al. (2016) [12].

### 2.3. Behavioural Traits

Animals were video recorded for 24 h post-mixing. Each animal that initiated or received an aggressive interaction was registered, including animal identity and type of interaction (fighting and/or bullying), as well as the corresponding duration. Fighting behaviour was defined as aggression that lasted at least one second where both pigs engaged in biting, pushing or head knocking the opponent. The bullying behaviour was defined as when one pig received or delivered aggression with no observable retaliation occurring [11].

### 2.4. SNA Traits

The SNA traits considered in this study were derived from Foister et al. (2018) [18] using the “igraph” package in R (version 3.2.3) [26,27]. The definitions and interpretations of the SNA traits are listed in Table 1. Briefly, SNA transforms the numerical data of aggressive interactions between animals into graphs, where the animals are displayed in terms of nodes. The interactions between animals are represented through edges, i.e., lines, that connect the nodes [28,29]. To include the different types of aggression occurred between animals in the analysis, the observed fighting and bullying interactions were combined and used as an input for the SNA. All of these were entered as undirected, allowing for bidirectional initiation of interactions, and unweighted, i.e., the frequency or duration of interactions between a given dyad were not considered, as both the weighted and unweighted networks have shown similar results for predicting pen lesions [18]. The SNA traits for each animal describe its position in the network, i.e., the pen. Here, we considered the SNA traits that were found to have a significant effect on the individual pig lesions based on the previous study of Foister et al. (2108) [18]. These include betweenness centrality, closeness centrality, degree centrality, eigenvector centrality, clustering coefficient as well as the binary trait clique membership defined in Table 1. All of the listed continuous centrality traits showed considerably skewed distributions; therefore, a square root transformation was applied to approach a normal distribution. The phenotypic associations between SNA traits were calculated using Spearman rank correlation using R software.

Whilst betweenness centrality is a continuous trait that assigns to each animal an individual betweenness centrality value, this trait was found to have a highly positively skewed distribution, as often only 1 or 2 individuals in each group have high betweenness centrality, whilst the majority of animals have very low or zero betweenness centrality. Therefore, we also transformed the betweenness centrality to a categorical trait, called “categorical betweenness” with two categories: ‘high’, comprising individuals within the top quartile of the betweenness centrality, and ‘low’, comprising individuals with the remaining 75%. The mean betweenness centrality of the top quartile was 0.13 (with a maximum of 0.4), and for the low betweenness category it was 0.01. Both the continuous and the categorical betweenness traits were included in the genetic analysis to investigate their genetic components.

### 2.5. Genetic Parameter Estimates

A series of univariate and bivariate analyses were used to estimate the genetic variance components and EBVs of all transformed SNA and skin lesion traits using the following linear animal model:y = Xb + Za + Wc + e
where y is the vector of records for the betweenness, closeness, degree, eigenvector centralities and clustering coefficient traits or anterior, central or posterior SL24h or SL3wk, respectively, and X, Z and W are the incidence matrices of fixed effects, random genetic effects and environmental (pen) effects, respectively. Vectors b, a, c and e represent fixed effects, additive direct genetic effects, common environmental pen effects and residual error, respectively. The fixed effects included the genetic line (2 levels; purebred Yorkshire and Yorkshire × Landrace), sex (3 levels; male, female, and castrated pigs) and batch, (14 levels; groups mixed on the same day were classed as the same batch), and the body weight at time of mixing was fitted as a covariate.

The categorical betweenness and clique membership traits were analysed using the threshold liability animal models with the same fixed and random effects as used in the linear animal model. In the threshold model an unobservable normally distributed variable, i.e., liability, is assumed for each observation. According to this, the observed categorical value is realized depending on whether the liability falls below or above a certain threshold value [30,31].

Bayesian analyses using the Markov Chain Monte Carlo (MCMC) method of Gibbs sampling were performed to estimate the models’ parameters using GibbsF90 from BLUPF90 family software [32] for the implementation of the linear animal model, while the software THRGIBBSF90 [33] was used for the implementation of the Threshold model. Flat priors were assumed for the systematic effects and the co-variance components. The conditional prior distributions of the random effects a, c and e were sampled from multivariate normal distributions (N) as follows:P(aǀA, G_0_) ~ N(0, A ⊗ G_0_)
P(cǀC_0_) ~ N(0, *I* ⊗ C_0_)
P(eǀR_0_) ~ N(0, *I* ⊗ R_0_)
where A is the additive genetic relationship matrix, G_0_ is the additive genetic (co)variance matrix and *I* is the identity matrix, and ⊗ represents the Kronecker product of the matrices. C_0_ is the (co)variance matrix of the environmental pen effects and R_0_ represents the (co)variance matrix of the residuals. The conditional posterior distributions of co-variance components of G_0_, C_0_ and R_0_ were sampled from inverse-Wishart distributions. The posterior distribution of the genetic and environmental effects was derived from a Markov Chain of 1,000,000 iterations of the parameters, where the burn in of 100,000 was applied to obtain a stationary MCMC distribution and a lag of 20 iterations was applied to reduce the autocorrelations between consecutive iterations. The convergence of the Markov Chain was tested using the algorithms of Raftery and Lewis (1992) [34]. Posterior means and the standard deviations of the marginal distributions of genetic and environmental parameters of SNA and lesion traits were estimated. Heritability, common environmental pen effect and genetic correlations between traits were calculated in addition to the 95% highest posterior density intervals (HPD95%) for each parameter to examine the credibility of the estimates.

### 2.6. Estimating the Effects of Selection for SNA Traits on Individual and Pen-Level Skin Lesions

As the SNA and skin lesion traits were measured on different scales, the EBVs obtained from the analyses were standardized into z-scores, i.e., expressed in terms of standard deviation with a mean of zero. Amongst the non-categorical traits, eigenvector centrality and clustering coefficient emerged as the two complementary SNA traits most closely and antagonistically related to the skin lesions at different body regions and time points in the above analyses. Therefore, to estimate the effect of selecting for these two SNA traits on skin lesions at the individual and pen level, the same method as in Desire et al. (2016) [12] was chosen, where only animals with SNA EBVs in the lowest 10% of the population were selected, and the corresponding mean EBVs of the various skin lesion traits were calculated. This method was chosen as it allows the prediction of selection response [12]. Additionally, it indirectly captures the effects resulting from the behaviour of pen mates on individuals’ estimated breeding values and phenotypes as outlined more in the discussion. Furthermore, we developed an index that includes both eigenvector centrality and clustering coefficient traits, i.e., eigenvector-clustering index, by summing the EBVs of both traits for each animal and considering that both have the same weight in the index. Then, to estimate the effect of selection for this index, animals with EBVs in the lowest 10% of the population were considered and the corresponding mean EBVs and phenotypes of the various skin lesion traits were calculated.

As selection for the eigenvector centrality and clustering coefficient traits would be expected to affect the pen’s network structure, therefore, both the average EBV as well as the standard deviation of EBVs at the pen-level, as a measure of diversity for these traits, were calculated. The pens were then ranked based on their average or standard deviation in the EBVs of these SNA traits, and the corresponding average skin lesions for the lowest 20% of pens was calculated.

## 3. Results

The descriptive statistics for the transformed SNA traits, skin lesion traits and the number of fights and bullying behaviours considered in this study are shown in Table 2. There was considerable phenotypic variation in all traits.

### 3.1. Heritability

The marginal posterior distributions and 95% highest posterior density intervals indicate that the heritability estimates for all SNA traits were significantly above zero (Table 3). The posterior means of heritability for SNA traits ranged from 0.09 for closeness centrality to 0.26 for both betweenness and degree centrality. The posterior means of the common environmental pen effect were lower than the heritability for all SNA traits, except for closeness centrality and clustering coefficient.

### 3.2. Genetic and Phenotypic Correlations

The posterior means of the genetic correlations and the HPD95% for SNA traits are shown in Table 4. SNA traits were strongly genetically correlated. High positive posterior means of genetic correlations were observed between the betweenness, closeness, degree, and eigenvector centrality traits (rg > 0.96). The corresponding phenotypic correlations between these traits were generally lower, but still moderate to strong (r > 0.53) (Appendix A). In contrast, the clustering coefficient showed high negative genetic correlations with these SNA centrality traits (rg < −0.88), although low and moderate negative correlations were observed between these traits at the phenotypic level. The genetic correlation between the two categorical traits, categorical betweenness and clique membership, was also high with HPD95% that did not include zero (rg = 0.86, HPD95% = 0.55, 0.99).

The posterior means of the genetic correlations and the HPD95% of SNA traits and skin lesions, recorded 24 h post-mixing, are presented in Table 5. Positive genetic correlations, with the HPD95% above zero, were observed between all SNA traits and anterior SL24h, except for clustering coefficient, which was weakly negatively correlated with anterior SL24h. Amongst all SNA traits, clique membership showed the strongest genetic correlation with anterior SL24h (rg = 0.96). The estimated genetic correlations between SNA traits and central and posterior SL24h were generally low with the HPD95% covering zero, except for moderate positive correlations with clustering coefficient. Conversely, estimated genetic correlations between SNA traits and anterior SL3wk were found to be negative, except for the clustering coefficient where the relationship was positive (Table 6).

### 3.3. Estimating the Effects of Selection for SNA Traits on Individual and Pen-Level Lesions

Animals selected to have EBVs in the lowest 10% for eigenvector centrality (EBVs mean = −2.48 SD) had lower than average EBVs for anterior SL24h (EBVs mean = −1.53 SD), while they had higher posterior SL24, anterior, central and posterior SL3wk (Figure 1a). In contrast, animals selected for EBVs in the lowest 10% for clustering coefficient (EBVs mean = −1.82 SD) had significantly lower EBVs than the population mean for all SL24h and SL3wk, except for the anterior SL24h. However, the largest reduction in skin lesion EBVs would be expected for the posterior SL3wk (EBVs mean = −1.10) (Figure 1b). Combining both SNA traits, animals with EBVs in the lowest 10% for eigenvector-clustering index (EBVs mean = −1.95 SD) revealed significantly lower EBVs than average for anterior and central SL24h, while the other skin lesions did not differ significantly from the population mean (Figure 1c). At the pen level, the lowest 20% of pens ranked based on the pen average EBVs for eigenvector centrality, clustering coefficient and the eigenvector-clustering index showed the same trend as the individual levels (Appendix A).

The phenotypic values for individuals with lowest 10% of EBVs for eigenvector centrality and clustering coefficient largely mirrored those observed on the genetic level for all skin lesions, except for posterior SL3wk for eigenvector centrality (Figure 2a), and central and posterior SL3wk for clustering coefficient (Figure 2b). However, the phenotypic values for individuals with the lowest 10% of EBVs for eigenvector-clustering index mirrored those observed on the genetic level for all skin lesions (Figure 2c).

The mean EBVs of the skin lesions in the lowest 20% of pens ranked based on the standard deviation of the EBVs for eigenvector centrality, clustering coefficient and eigenvector-clustering index are shown in Figure 3. Pens with low variation in EBVs for eigenvector centrality and clustering coefficient had lower SL24h and SL3wk than average at different parts of the body, except for anterior SL24h for both SNA traits, and additionally for central SL24h for clustering coefficient. On the other hand, the mean EBVs of the skin lesions in the lowest 20% of pens ranked based on the standard deviation of the EBVs for eigenvector-clustering index had low levels of SL24h and SL3wk at different parts of the body, except for anterior skin lesions at the two time points.

## 4. Discussion

### 4.1. Heritability

Few studies have explored the genetic determination underlying SNA traits. Here, the posterior means of the heritability for the SNA traits of aggressive behaviour were low to moderate. To the best of our knowledge, the heritability of SNA traits in pigs have not been previously estimated. However, the heritability estimates found here, except for closeness centrality, are similar to the heritability reported for SNA of aggressive behaviour in different species (h^2^ ranged from 0.11 to 0.66) [22,23,24,25]. Furthermore, the heritability estimates for all SNA traits are within the range of heritability estimates of dyadic behavioural traits in pigs (h^2^ = 0.04–0.43) [12,35], and skin lesions at 24 h post-mixing (h^2^ = 0.11–0.43) [8,9,12,36]. It is remarkable that the heritability estimates for the SNA traits are similarly high as for the dyadic or skin lesion traits, whilst they also capture part of the interactions between other individuals in the group other than the subject itself. The magnitude of the heritability of SNA traits indicates that these traits are partially under genetic regulation and could be utilized for selective breeding.

Closeness centrality showed the lowest heritability estimates among the SNA traits (h^2^ = 0.09). This trait measures how close an animal is to all other animals in the network, which reflects the degrees of separation or steps between individuals [28]. Thus, amongst the SNA traits considered here, an individual’s closeness centrality most strongly depends on the structure of aggressive interactions within the pen. This may explain the relatively low direct genetic effect contributing to the genetic variation of this trait and the high environmental pen effect compared to other centrality traits (c^2^ = 0.59).

### 4.2. Genetic and Phenotypic Correlations

By definition, centrality traits are expected to be both genetically and phenotypically correlated as they are not independent observations (Table 1 and Appendix A). Genetic correlations were generally in the same direction but stronger than phenotypic correlations, i.e., close to unity. Removal of the pen effect in the statistical models did not affect the genetic correlation estimates, except for closeness for which the genetic correlation with other SNA traits was reduced by removing the pen effect. This suggests that the observed high genetic correlations are genuine, i.e., at the genetic level, the diverse SNA traits are almost identical, rather than an artefact of the adjustment of the SNA traits for the pen level.

Positive genetic correlations were found between degree, betweenness, closeness, eigenvector centralities and anterior SL24h (rg > 0.38). In contrast, these SNA centrality traits were generally negatively genetically correlated with SL3wk on different body regions. That would suggest that individuals with a genetic predisposition for high centrality immediately after mixing would tend to suffer high injuries on the anterior part of the body at that stage but would tend to have lower injuries in the stable group. A similar trade-off has been previously observed for dyadic aggressive interactions, where reciprocal aggression, in which the recipient of the attack retaliated, showed strong and positive genetic correlations with anterior SL24h (rg > 0.75) and negative correlations with skin lesions in the stable group [12]. Thus, combining these results suggests that animals with a genetic predisposition for a central location in the social network where they engage in aggressive behaviour with opponents who themselves interact with several pen mates, tend to be more prone to receipt of a high number of lesions directly after mixing but few lesions in the stable group. This is particularly true for clique membership, which showed the highest positive genetic correlation, among the SNA traits, with anterior SL24h, and a strong negative genetic correlation with anterior SL3wk. This is in line with the results reported previously, at the phenotypic level, which demonstrated that the individuals belonging to the largest clique in their pen received significantly more anterior SL24h compared to the non-clique members, and lower injuries in the stable groups [18]. This would indicate that individuals that are genetically prone towards clique membership establish their hierarchical position through aggressive interactions, resulting in receipt of injury after mixing, but this decreases injuries from involvement in long term aggression.

Amongst all SNA traits, the clustering coefficient was the only trait that was found to be positively genetically correlated with skin lesions at both time points (i.e., central and posterior SL24h, and anterior and central SL3wk). This would suggest that using the clustering coefficient at mixing as selection criterion may not inflict a trade-off for reducing lesions at different time points or different body parts. However, it also needs to be considered that the clustering coefficient is strongly negatively correlated with all other SNA traits.

Except for the clustering coefficient, the genetic correlations between SNA traits with central and posterior SL24h were generally low, with high levels of uncertainty and the 95% highest posterior density intervals spanning a wide range of values both below and above zero. In line with these results, Desire et al. (2016) [12] found no significant genetic correlations between central SL24h and other behavioural traits recorded at the dyadic level. The central lesion trait was found to be an ambiguous proxy of aggression, as it could capture both aggressive and non-aggressive animals [37]. That may partly explain the relatively high variation in the genetic correlation estimates found between central skin lesions and SNA traits.

### 4.3. Estimating the Effects of Selection for SNA Traits on Individual and Pen-Level Lesions

Although genetic correlations are valuable parameters to understand the genetic relationship among traits, the estimation of genetic correlations typically has a high level of uncertainty and requires substantial amounts of data [38]. Furthermore, both SNA and skin lesion traits are strongly influenced by the aggressive behaviour of other individuals. Therefore, in this study, the prediction of the effect of selection for SNA traits on skin lesions was based on the EBVs of the individuals which allows more robust prediction compared to the genetic correlations [39] and reflects a potential correlated selection responses. This approach was also previously used to predict the reduction in dyadic aggressive behaviour traits when using skin lesions as the criteria of selection in pigs [12]. Given the high genetic correlations between all SNA traits, we focused on eigenvector centrality and clustering coefficient as these two traits capture both individual’s direct engagement in aggressive interactions as well as that of their pen mates. In addition, these traits were found to be strongly and antagonistically related to skin lesions at different body regions 24 h post-mixing and in the stable group.

Consistent with the strength and direction of genetic correlation estimates, our results showed that animals with low EBVs for eigenvector centrality had low genetic and phenotypic values for anterior SL24h. However, it should be noted that these animals showed high EBVs for the posterior SL24h and SL3wk on all parts of the body (Figure 1a); however, these associations were not universally observed on the phenotypic level (Figure 2a). These findings suggest that selection for low eigenvector centrality would benefit the group in the short term, as it would decrease aggression and injuries 24 h post-mixing but could increase the injuries in the stable group conditions. Foister et al. (2018) [18] found that pens containing few animals with high eigenvector centrality were significantly associated with on-going aggression and injury in the pen at 3 weeks post mixing. On the other hand, the animals with EBVs in the lowest 10% for clustering coefficient had low EBVs for all SL24h and SL3wk, except for anterior SL24h (Figure 1b). Thus, selection for decreasing clustering coefficient would be expected to decrease injuries in central and posterior body regions at the two time points, although it could increase the anterior injuries immediately after mixing. These results would indicate that there is a trade-off when considering the eigenvector centrality and clustering coefficient as criteria of selection regarding their effect on skin lesions. Therefore, we suggested here an index that combines both traits, i.e., eigenvector-clustering index, and we predicted the effect of using this index as criteria of selection on skin lesion traits. Our results showed that animals in the lowest 10% of EBVs for eigenvector-clustering index had low EBVs for skin lesions in all body parts at both time points, except the central and posterior SL3wk which were not significantly different from the population mean (Figure 1c). Furthermore, the phenotypic values of skin lesions for these individuals mirrored those observed at the genetic level (Figure 2c). These findings suggest that selection for the eigenvector-clustering index would potentially decrease the injuries and benefit the group in the short-term and long-term compared to selection for eigenvector centrality and clustering coefficient separately.

At the pen level, the lowest 20% of pens ranked based on the pen average EBVs for eigenvector centrality, clustering coefficient and the eigenvector-clustering index showed the same trend as the individual level (Appendix A). Thus, selection on eigenvector centrality or clustering coefficient of each individual pig would be expected to result in similar correlated changes in skin lesion traits at the pen level. This result may reflect the fact that these SNA traits capture not only the characteristics of individuals but also the social interactions among all pigs in the pen, associated with an overall reduction in skin lesions at pen level.

Selection for SNA traits could lead to a change in the structure of the network, i.e., the pen, as it could lead to a reduction in the variation in these traits. Therefore, we also investigated the effect of the variation in eigenvector centrality and clustering coefficient traits within the pen on skin lesions. Our findings suggest that the pens with low variation in EBVs for eigenvector centrality and clustering coefficient would have low SL24h and SL3wk at different parts of the body, except anterior SL24h in both traits, and central SL24h for clustering coefficient (Figure 3a,b). On the other hand, low variation in the pen level EBVs for eigenvector-clustering index showed low pen level skin lesions in all body parts at the two time points (Figure 3c). Thus, aiming to reduce variation in the eigenvector-clustering index through selection would help in decreasing the average injuries in the pen compared to reducing variation in eigenvector centrality and clustering coefficient separately.

The latter results would also suggest that the reduction in the mean index was more important for reducing anterior skin lesions than the index variation around the mean pen level. However, it is necessary to emphasise that selection mainly on the eigenvector-clustering index and secondary on the variation of this index within pen could be used to achieve a reduction in central and posterior SL3wk in the stable group. Moreover, the weighting of the SNA traits within the index could be changed to achieve different correlated responses in lesion traits, as both eigenvector centrality and clustering coefficient are inversely affecting skin lesion traits.

### 4.4. Future Prospects for Breeding against Aggressive Behaviour in Pigs

Breeding against aggressive behaviour in pigs is challenging. The determination of the appropriate traits to reduce the pen level aggression and skin lesions and choosing which traits are easy to measure in commercial farms are important aspects for improving the welfare of pigs. SNA has shown the potential in providing measures that describe the direct and indirect social relationships between farm animals [17,19]. Advances in automated capture and analysis of animal behaviour will facilitate the application of SNA in the breeding industry [40]. Furthermore, the genetic parameters of the SNA traits estimated in this study indicate that these traits are amenable for selective breeding.

However, this study only provides a first step towards potential integration of SNA traits into selection strategies for improving animal welfare by reducing both aggressive interactions as well as resulting injuries. Antagonistic genetic relationships between the diverse SNA traits and skin lesions in different body regions obtained at mixing versus stable groups point towards potential trade-offs between reducing aggression and resulting injuries at different body regions and time points when using SNA traits separately as criteria of selection. However, the effect of this trade-off can be decreased by considering the eigenvector-clustering index, suggested in this study, although further studies are needed to confirm this, and also to better understand the genetic influence underlying group level aggression in different group compositions, environments and production systems.

Compared to dyadic behavioural traits, SNA traits describe an individual’s direct engagement in aggressive behaviour in the context of the group’s behavioural structure defined by social interactions between all group members. Previous studies have highlighted that the social genetic effects play an important role in skin lesions resulting from aggressive interactions of pigs at mixing [41,42]. Furthermore, estimates of correlations between direct and social genetic effects for skin lesions, as well as dyadic aggressive behaviour were found to be positive [43,44] implying that selecting animals with low genetic propensity for engaging in aggressive behaviour or for receiving skin lesions may be beneficial for reducing aggression and skin lesions in the group as a whole. However, estimation of social genetic effects requires very large data and a particular data structure that may be difficult to obtain in commercial settings [45]. In contrast, selection for SNA traits, which show similar heritability estimates as dyadic behavioural traits but intrinsically incorporate social interactions, may thus be a more efficient way to reduce aggressive behaviour and resulting skin lesions at the pen level, particularly as the latter were shown to be affected by both changes in the mean as well as in the intra-pen variation in SNA traits.

Lastly, SNA traits may enable selection for more socially tolerant or socially skilled pigs that contribute to lower pen level aggression and injuries, thus maintaining high performance in the social environments that prevail on commercial farms. Such selection should be complemented by continued efforts to find commercially feasible management interventions to reduce aggression. However, it is important to consider the association between these behaviour measures and economically important traits, e.g., performance and feeding behaviour traits [19].

## 5. Conclusions

Social network analysis could be considered as a promising approach to describe the social interactions underlying harmful aggressive behaviour of animal groups. The results of this study indicate that many relevant SNA traits are moderately heritable and strongly genetically correlated but have different genetic correlations on the outcome of aggressions identified as skin lesions. Thus, the genetic correlations between SNA traits and skin lesions, and EBVs based estimates of selection response at the individual and pen level, suggest that selecting for different SNA traits separately would have different effects on skin lesions at different body parts immediately after mixing and in stable groups. Selection for an index that combines the eigenvector centrality and clustering coefficient could potentially decrease skin lesions after mixing and in the stable group environment at different parts of the body. Incorporating the SNA and other behaviour traits, along with the economically important traits is recommended for establishing a future strategy for simultaneously improving the performance and welfare of pigs.

## Figures and Tables

**Figure 1 genes-13-00561-f001:**
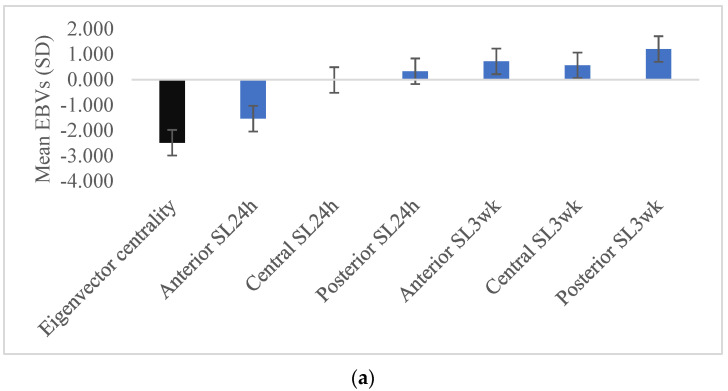
Mean estimated breeding values (EBVs) and standard errors of skin lesions traits of pigs with EBVs in the lowest 10% for eigenvector centrality (**a**), clustering coefficient (**b**) and eigenvector-clustering index (**c**). The trait that selection was based on is shaded black.

**Figure 2 genes-13-00561-f002:**
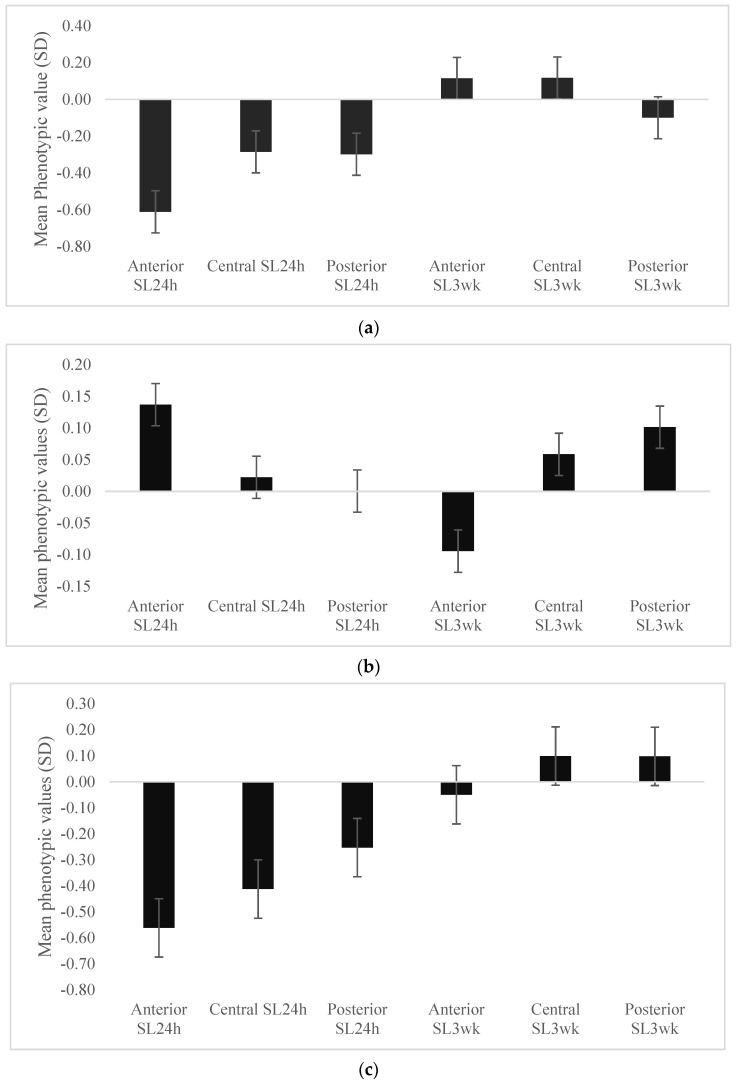
Mean phenotypic values and standard errors of skin lesions traits of pigs with estimated breeding values in the lowest 10% for eigenvector centrality (**a**) and clustering coefficient (**b**) and eigenvector-clustering index (**c**).

**Figure 3 genes-13-00561-f003:**
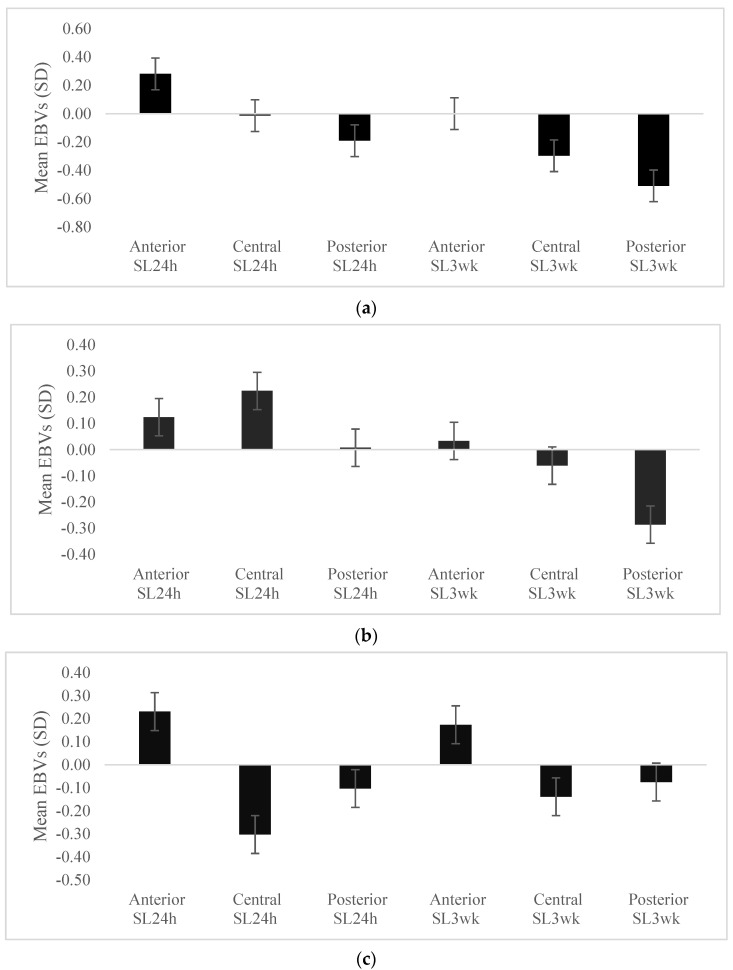
Mean estimated breeding values (EBVs) and standard errors of the skin lesions for the lowest 20% of pens ranked based on the standard deviation of the EBVs for eigenvector centrality (**a**) and clustering coefficient (**b**) and eigenvector-clustering index (**c**).

**Table 1 genes-13-00561-t001:** The definition of the social network analysis traits considered in this study.

Measures	Definition	Interpretation
All-degree centrality	The number of edges attached to a node.	The number of animals that a particular animal directly engaged with.
Betweenness centrality	The number of shortest paths that pass through the considered node.	Measures the importance of the animal in connecting different subgroups of the pen engaging in aggression.
Categorical betweenness	A transformation of the betweenness centrality to a binary trait that has two categories.	Individuals within the top quartile of the betweenness centrality are considered ‘high’, whereas individuals in the remaining 75% are considered as ‘low’.
Closeness centrality	The average of the shortest path length between that node and all other nodes in the network.	Measures how ‘close’ an animal is to all other animals in a pen in terms of engaging in aggression. Animals that engage in aggression directly with many of their pen mates have high closeness centrality.
Eigenvector centrality	The connectivity of a node within its network, according to the all-degree centrality of the node and the all-degree centrality of the nodes that it connects with.	Takes into consideration both the number of aggressive interactions of the focal individual and the number of aggressive interactions that its social partners have.
Clustering coefficient	The proportion of an individual node’s connections that are also connected with each other relative to the possible number of theoretically possible connections.	Quantifies what proportion of animals that the focal individual directly engages with also interact with each other, relative to the number of all possible aggressive interactions.
Clique membership	A categorical trait where individuals are categorized based on whether or not they are members of the largest clique(s) in the group (‘clique members’ or ‘non-clique members’).	A clique is a fully connected subgroup of animals in a pen, where each animal engages in aggressive interactions with every other animal in that sub-group.

**Table 2 genes-13-00561-t002:** Descriptive statistics for the transformed aggressive social network traits, skin lesion traits and the number of fights and bullying behaviours.

Trait	Mean ± SD	Maximum	Minimum
Betweenness centrality	0.04 ± 0.05	0.44	0
Closeness centrality	0.48 ± 0.09	0.69	0.06
Degree centrality	0.38 ± 0.15	0.69	0
Eigenvector centrality	0.21 ± 0.07	0.39	0
Clustering coefficient	0.46 ± 0.15	0.69	0
Clique membership (size of large clique)	-	8	4
anterior SL24h	2.57 ± 1.08	4.61	0
central SL24h	2.05 ± 1.10	4.62	0
posterior SL24h	1.36 ± 1.02	3.74	0
anterior SL3wk	2.30 ± 0.57	4.16	0
central SL3wk	2.27 ± 0.60	3.71	0
posterior SL3wk	1.48 ± 0.71	3.43	0
Number of fights initiated	4.1 ± 4.3	35	0
Number of fights received	4.2 ± 3.8	24	0
Number of bullying initiated	3.8 ± 5.8	66	0
Number of bullying received	3.7 ± 5.8	25	0

**Table 3 genes-13-00561-t003:** Posterior means of heritability (h^2^), the phenotypic proportions of the variance due to the environmental pen effects (c^2^), and the phenotypic variances (Vp), and the 95% highest posterior density intervals (HPD95%) for social network analysis traits for aggressive behaviour.

Trait	h^2^	HPD95%		c^2^	HPD95%		Vp	HPD95%	
Betweenness centrality	0.26	0.11	0.40	0.02	0.003	0.03	0.015	0.014	0.017
Categorical betweenness	0.15	0.02	0.29	0.05	0.00	0.11	1.644	1.043	2.47
Closeness centrality	0.09	0.04	0.17	0.59	0.49	0.69	0.006	0.005	0.008
Degree centrality	0.26	0.12	0.41	0.14	0.08	0.21	0.010	0.009	0.011
Eigenvector centrality	0.22	0.11	0.36	0.01	0.001	0.02	0.006	0.006	0.007
Clustering coefficient	0.18	0.07	0.29	0.23	0.15	0.31	0.008	0.007	0.009
Clique membership	0.18	0.06	0.35	0.11	0.02	0.20	1.51	1.16	1.96

**Table 4 genes-13-00561-t004:** Posterior means of the genetic correlations and the 95% highest posterior density intervals (HPD95%) for social network analysis traits for aggressive behaviour considered in this study.

Trait	Closeness Centrality	HPD95%		Degree Centrality	HPD95%		Eigenvector Centrality	HPD95%		Clustering Coefficient	HPD95%	
Betweenness centrality	0.97	0.93	0.99	0.99	0.96	1	0.96	0.94	1	−0.95	−0.99	−0.90
Closeness centrality				0.98	0.97	1	0.97	0.94	0.99	−0.98	−1	−0.92
Degree centrality							0.98	0.96	0.99	−0.88	−0.99	−0.72
Eigenvector centrality										−0.95	−0.99	−0.85

**Table 5 genes-13-00561-t005:** Posterior means of the genetic correlations and the 95% highest posterior density intervals (HPD95%) of social network analysis traits for aggressive behaviour and skin lesions recorded 24 h post-mixing (SL24h).

Trait	Anterior SL24h	HPD95%		Central SL24h	HPD95%		Posterior SL24h	HPD95%	
Betweenness centrality	0.46	−0.01	0.87	−0.10	−0.71	0.40	−0.27	−1.00	0.41
Categorical betweenness	0.38	−0.07	0.89	−0.27	−1.00	0.53	−0.11	−0.97	0.49
Closeness centrality	0.49	0.13	0.82	0.10	−0.46	0.50	−0.23	−0.99	0.40
Degree centrality	0.62	0.34	0.88	0.01	−0.46	0.44	−0.02	−0.60	0.56
Eigenvector centrality	0.54	0.15	1.00	−0.19	−1.00	0.39	−0.13	−1.00	0.59
Clustering coefficient	−0.14	−0.56	0.34	0.63	0.08	1.00	0.73	0.24	1.00
Clique membership	0.96	0.84	1.00	−0.57	−1.00	0.93	0.36	−0.23	0.95

**Table 6 genes-13-00561-t006:** Posterior means of the genetic correlations and the 95% highest posterior density intervals (HPD95%) of social network analysis traits recorded within 24 h post-mixing and skin lesions recorded 3 weeks post-mixing (SL3wk).

Trait	Anterior SL3wk	HPD95%		Central SL3wk	HPD95%		Posterior SL3wk	HPD95%	
Betweenness centrality	−0.57	−0.99	−0.24	−0.49	−0.87	−0.09	−0.78	−1.00	0.18
Categorical betweenness	−0.37	−0.86	0.06	0.01	−0.71	0.59	−0.20	−0.85	0.51
Closeness centrality	−0.33	−0.67	0.01	−0.28	−0.61	0.09	−0.06	−0.96	0.72
Degree centrality	−0.37	−0.66	−0.06	−0.18	−0.56	0.20	0.06	−0.53	0.80
Eigenvector centrality	−0.47	−0.84	−0.10	−0.23	−0.67	0.19	−0.17	−0.76	0.40
Clustering coefficient	0.68	0.28	1.00	0.62	0.26	0.97	0.40	−0.22	0.98
Clique membership	−0.84	−1.00	−0.40	−0.57	−1.00	0.15	−0.46	−0.99	0.46

## Data Availability

Data can be made available upon request.

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
