# Peer review of "Genetic Analysis of Novel Behaviour Traits in Pigs Derived from Social Network Analysis"

_genes, 2022, doi:10.3390/genes13040561_

Round 1

Reviewer 1 Report

The objectives of this study were to: estimate the genetic parameters for these SNA traits, quantify the genetic association between SNA and skin lesion traits, and investigate the possible response to selection for SNA traits on skin lesion traits. The results of this study showed possibility to improving the welfare of pigs. 
I have some minor comments for the M&M section. 
It is necessary to briefly describe the technology and breeding system: the pens are in one building or are divided into departments/chambers (number of animals); the layout of the pens in the facility; microclimate control; floor; feeders and the number of feeding places per pen.....

Author Response

Cover letter

First, we would like to thank the reviewers for their valuable comments. Please find below our reply on each comment in “blue”.

  • The objectives of this study were to: estimate the genetic parameters for these SNA traits, quantify the genetic association between SNA and skin lesion traits, and investigate the possible response to selection for SNA traits on skin lesion traits. The results of this study showed possibility to improving the welfare of pigs. 
    I have some minor comments for the M&M section. 
    It is necessary to briefly describe the technology and breeding system: the pens are in one building or are divided into departments/chambers (number of animals); the layout of the pens in the facility; microclimate control; floor; feeders and the number of feeding places per pen.....
  • We added the following sentence in Line 91 as the reviewer suggested.

“Briefly, the dataset consisted of 1146 commercial pigs, 698 of which were purebred Yorkshire and 448 were Yorkshire × Landrace on a commercial farm in Ransta, Sweden, between October 2005 and January 2007. They were housed in the same building with each ‘batch’ of animals (i.e. those on the experiment at the same moment in time) housed in the same room of this building. The building had automatically controlled natural ventilation with the temperature maned within the thermoneutral zone and adjusted as the pigs aged. Pigs were kept in 4.0 × 3.2 m partially slatted pens (30% slats, 70% lightly bedded solid flooring) with a floor space allowance of 0.85 m2 per pig. Animals were fed dry pelleted food ad libitum from a single space feeder and had constant access to water via a nipple drinker.”

Reviewer 2 Report

In this manuscript, Agha et al. used social network analysis (SNA) to analyze the aggressive behavior traits of pigs, analyzed the genetic parameters of SNA traits and the genetic correlation between SNA traits and skin lesion traits, and finally evaluated the feasibility of SNA traits on the selection of skin lesion traits. The authors found that SNA traits are heritable and had strong genetic correlation. All SNA traits were positively correlated with the skin lesion score at 24 h post-mixing. The selection index considering the eigenvector centrality and clustering coefficient could reduce the skin lesion and improve the welfare of pigs. In general, this paper is very interesting.

Line 39-52: There are still different views on whether skin lesions can be used as an indicator of individual selection in pig breeding. Important recent reference is missing here: "Liu et al. 2022. Pigs' skin lesions at weaning are primarily caused by standoff and being bullied instead of unilateral active attack at the individual level. Applied Animal Behavior Science".

Line 56-62: There should be some relevant references cited here. For example, "Tong et al. 2019. Reestablishment of Social Hierarchies in Weaned Pigs after Mixing".

Line 100-107: How to distinguish skin lesions of different lengths, depths and widths? Can two scratches be distinguished if they are next to each other? How does bleeding lesion and do not bleed lesion distinguish? Without detailed information about skin lesion traits, whether the lesion data was a good indicator of the extent of the injury concerns reviewer.

Line 108-114: How about the behavior of standoff: two pigs stand side by side, shoulder by shoulder, head knocking? How to know which one is "initiated" or "received"?

The goal of animal breeding is to reduce the aggression of animals. But the two fighting pigs were not both aggressive. Maybe one of them was forced to fight (self-defense). Therefore, as the analysis of aggressive behavior for animal breeding, it is more meaningful to focus on active (initiated) aggressive behavior. If we did not distinguish between active aggressive behavior and passive self-defense aggressive behavior, the assessment of aggressiveness of pigs may be illogical.

Line 130 There is an error in the format of the citation, "2108" should be changed to "2018”.

Table 2: The skin lesions data in the table showed no difference between skin lesions at 24 hours and 3 weeks later after mixing, indicating that the aggressive behavior of pigs did not increase after mixing. Why?

In addition, why is there no descriptive statistics for behavior parameters (the initiated or received aggressive fighting and bullying) listed in Table 2? I would like to see descriptive statistics of the raw behavioral data in Table 2.

I suggest that Figure 1 and Figure 2 can be combined together.

Author Response

Cover letter

First, we would like to thank the reviewers for their valuable comments. Please find below our reply on each comment in “blue”.

In this manuscript, Agha et al. used social network analysis (SNA) to analyze the aggressive behavior traits of pigs, analyzed the genetic parameters of SNA traits and the genetic correlation between SNA traits and skin lesion traits, and finally evaluated the feasibility of SNA traits on the selection of skin lesion traits. The authors found that SNA traits are heritable and had strong genetic correlation. All SNA traits were positively correlated with the skin lesion score at 24 h post-mixing. The selection index considering the eigenvector centrality and clustering coefficient could reduce the skin lesion and improve the welfare of pigs. In general, this paper is very interesting.

Line 39-52: There are still different views on whether skin lesions can be used as an indicator of individual selection in pig breeding. Important recent reference is missing here: "Liu et al. 2022. Pigs' skin lesions at weaning are primarily caused by standoff and being bullied instead of unilateral active attack at the individual level. Applied Animal Behavior Science".

  • Thanks for the suggestion. We added the reference (Reference number 10).

Line 56-62: There should be some relevant references cited here. For example, "Tong et al. 2019. Reestablishment of Social Hierarchies in Weaned Pigs after Mixing".

  • Thanks for the suggestion. We added the reference (Reference number 14).

Line 100-107: How to distinguish skin lesions of different lengths, depths and widths? Can two scratches be distinguished if they are next to each other? How does bleeding lesion and do not bleed lesion distinguish? Without detailed information about skin lesion traits, whether the lesion data was a good indicator of the extent of the injury concerns reviewer.

Previous studies have shown the phenotypic and genetic level that skin lesion number is correlated with involvement in aggressive behaviour (Desire et al., 2016; Turner et al., 2008, 2009). It should be noted that counting the absolute number of skin lesions is substantially more sensitive than categorising lesion number as often performed (e.g. in the EU Welfare Quality protocol). Skin lesions were counted at 24 hours post-mixing because our previous studies showed that inflammation of surrounding skin had subsided by this time. This allowed differentiation of adjacent lesions which would have been harder if counting had occurred earlier. Lesions were superficial and rarely deep enough to cause bleeding.  A lesion was recorded if it was vivid red in colour or recently scabbed. Typically, lesions were of the same width (generally only 1-2mm wide) and recording width of lesions would not yield additional information. We acknowledge that lesion length can vary, but recording length of lesions would not be feasible without prolonged and stressful restraint of the animals and would be laborious given the number of lesions per animal and number of animals in the population.

We added 2 sentences regarding this point in the paper in Lines 104 and 111.

Line 108-114: How about the behavior of standoff: two pigs stand side by side, shoulder by shoulder, head knocking? How to know which one is "initiated" or "received"?

The goal of animal breeding is to reduce the aggression of animals. But the two fighting pigs were not both aggressive. Maybe one of them was forced to fight (self-defense). Therefore, as the analysis of aggressive behavior for animal breeding, it is more meaningful to focus on active (initiated) aggressive behavior. If we did not distinguish between active aggressive behavior and passive self-defense aggressive behavior, the assessment of aggressiveness of pigs may be illogical.

  • The initiator was the animal that performed the first aggressive act (as defined in the paper). Where both animals initiated simultaneously, no initiator ID was recorded. Pigs have two choices when attacked – retaliate or run away. At this stage aggression is about establishing dominance. Retaliation is shown by a pig that believes it can use aggression and escalate in step with the initiator to secure a higher dominance position. Pigs that show retaliatory or responsive aggression are perhaps less of a problem than those which initiate, but that does not necessary mean they are unaggressive, or they are forced into defensive aggression due to the lack of an ability to withdraw.
  • However, we have already done the analyses for the fighting and bullying separately and the results were overall similar. We provide below 2 tables that show that the heritability estimates of the separate fighting and bullying interactions were similar (Please see the following tables). As our objectives were to reduce all types of aggressive interaction, we presented in this paper the results of the combined fighting and bullying interaction.

Table. Posterior means of heritability (h2), the phenotypic proportions of the variance due to the environmental pen effects (c2), and the phenotypic variances (Vp), and the 95% highest posterior density intervals (HPD95%) of social network analysis for the fighting behaviour.

Trait

h2

HPD95%

c2

HPD95%

Vp

      HPD95%

Betweenness Centrality

0.24

0.11

0.39

0.01

0.00

0.04

0.02

0.01

0.02

Closeness centrality

0.02

0.01

0.03

0.93

0.91

0.95

0.01

0.01

0.01

Degree centrality

0.24

0.11

0.40

0.17

0.09

0.26

0.01

0.01

0.01

Eigenvector centrality

0.18

0.06

0.32

0.02

0.001

0.05

0.01

0.01

0.01

Clustering coefficient

0.13

0.05

0.24

0.24

0.14

0.33

0.01

0.01

0.01

Table. Posterior means of heritability (h2), the phenotypic proportions of the variance due to the environmental pen effects (c2), and the phenotypic variances (Vp), and the 95% highest posterior density intervals (HPD95%) of social network analysis for the bullying behaviour.

Trait

h2

    HPD95%

c2

     HPD95%

vp

HPD95%

Betweenness Centrality

0.10

0.02

0.20

0.03

0.001

0.05

0.02

0.02

0.03

Closeness centrality

0.02

0.002

0.04

0.85

0.80

0.90

0.01

0.01

0.01

Degree centrality

0.12

0.04

0.21

0.20

0.12

0.28

0.02

0.01

0.02

Eigenvector centrality

0.10

0.03

0.18

0.01

0.00

0.03

0.01

0.01

0.01

Clustering coefficient

0.10

0.03

0.19

0.23

0.15

0.33

0.02

0.01

0.02

Line 130 There is an error in the format of the citation, "2108" should be changed to "2018”.

  • We changed the date to 2018.

Table 2: The skin lesions data in the table showed no difference between skin lesions at 24 hours and 3 weeks later after mixing, indicating that the aggressive behavior of pigs did not increase after mixing. Why?

  • The table shows the data in log transformed. The means on the untransformed scale are substantially different between these two time points. We added that to the table title to clarify this point for the reader.

In addition, why is there no descriptive statistics for behavior parameters (the initiated or received aggressive fighting and bullying) listed in Table 2? I would like to see descriptive statistics of the raw behavioral data in Table 2.

  • We added this information to Table 2 as the reviewer suggested.

I suggest that Figure 1 and Figure 2 can be combined together.

  • Regarding combining Figures 1 and 2, we would like to thank the reviewer for his suggestion. However, we don’t see much gain in combining them as these figures relate to different aspects (individual vs pen effects) and separating these figures may help to make it clearer for the reader.
